# Transcriptional Regulators of Plant Adaptation to Heat Stress

**DOI:** 10.3390/ijms241713297

**Published:** 2023-08-27

**Authors:** Xuejing Wang, Nicholas Wui Kiat Tan, Fong Yi Chung, Nobutoshi Yamaguchi, Eng-Seng Gan, Toshiro Ito

**Affiliations:** 1Department of Biological Science, Graduate School of Science and Technology, Nara Institute of Science and Technology, Ikoma 630-0192, Nara, Japan; wang.xuejing.xa0@bs.naist.jp (X.W.); nobuy@bs.naist.jp (N.Y.); 2School of Applied Science, Republic Polytechnic, Singapore 738964, Singapore; nicholas_tan_wk@rp.edu.sg (N.W.K.T.); chung_fong_yi@rp.edu.sg (F.Y.C.)

**Keywords:** heat stress, transcription factor, epigenetics, histone modification, protein homeostasis, ROS homeostasis

## Abstract

Heat stress (HS) is becoming an increasingly large problem for food security as global warming progresses. As sessile species, plants have evolved different mechanisms to cope with the disruption of cellular homeostasis, which can impede plant growth and development. Here, we summarize the mechanisms underlying transcriptional regulation mediated by transcription factors, epigenetic regulators, and regulatory RNAs in response to HS. Additionally, cellular activities for adaptation to HS are discussed, including maintenance of protein homeostasis through protein quality control machinery, and autophagy, as well as the regulation of ROS homeostasis via a ROS-scavenging system. Plant cells harmoniously regulate their activities to adapt to unfavorable environments. Lastly, we will discuss perspectives on future studies for improving urban agriculture by increasing crop resilience to HS.

## 1. Introduction

### 1.1. Global Warming and Its Impact on Plants

Climate change describes the gradual shift in average temperature and weather patterns over time, which is occurring worldwide. Such a phenomenon is primarily attributed to human activities, with burning fossil fuels and deforestation being the top two causal factors contributing to greenhouse gas emissions. The severity of heat stress due to global warming is expected to bring about catastrophic disruptions to the world food supply, with tropical regions experiencing the most significant rise in temperature over time [1]. Climate change is also predicted to impact agricultural productivity, including staple crops such as maize and wheat [2]. Climate models forecast that up to ten times the number of crops will be damaged by the end of the century due to rising global temperatures [3].

Elevated temperature causes heat stress (HS), an abiotic stress to plant growth and development, thereby influencing crop yield. In addition to perturbation of plant growth and development, HS impacts biological processes and metabolic pathways, including enzymatic activity, protein folding, and lipid oxidation. Some consequences include poor seed germination, stunted growth, inability to set fruits, and plant death [4]. Studies that focus on crop breeding for high-yielding varieties might compromise their tolerance to biotic and abiotic stresses. When exposed to HS, crop plants are unable to thrive, thereby impacting crop production and food security [4].

### 1.2. Thermotolerance in Plants

Despite the detrimental effects of HS, most plants are capable of adapting to moderate heat. According to the HS temperature regimes and developmental stages of the plants examined, there are various types of thermotolerance in plants that can be attained through inherent resistance and acquired thermotolerance (AT) [5,6]. There are typically three categories of organismal thermotolerance in response to various HS events in *Arabidopsis*, comprising basal thermotolerance to HS at temperatures ranging from 40 to 45 °C, AT following a short period of nonlethal temperature or priming HS, and thermotolerance to prolonged exposure to moderately high temperatures between 30 and 38 °C [5]. A series of molecular and cellular responses is triggered upon sensing heat stimulus. Mechanistic differences as well as overlapping but distinct molecular and cellular responses exist among these three modes of thermotolerance [5]. Crosstalk among transcription factors (TFs), such as heat shock factors or heat stress transcription factors (HSFs), regulatory RNAs, and epigenetic regulators, induces physiological responses during HS [6]. Some of the responses are reflected in the rate of plant photosynthesis, alteration of thermostability of the cell membrane, regulation of flowering time, and production of antioxidants for protection against oxidative stress [6].

Generally, plants adapt to HS via HSF- and heat shock protein (HSP)-mediated heat stress responses (HSRs). Understanding the underlying mechanisms of plant cell activities in response to HS, especially transcriptional regulation activity, is essential for safeguarding crop resilience and the development of heat-tolerant crop varieties. In this paper, the molecular mechanisms as a result of HS in plants will be reviewed, focusing on TFs.

## 2. Transcriptional and Epigenetic Regulation for Adaptation to HS

### 2.1. Plant Heat Stress Transcription Factors: Classes A, B, and C

Transcriptional regulation of genes involved in plant heat stress responses is guarded by HSFs (Figure 1). Plant HSFs, consisting of Classes A, B and C, share evolutionarily conserved core transcription regulators. The diversity and functions of HSFs were reviewed by Koskull-Doring et al. [7]. The functional domains of HSFs consist of a DNA-binding domain at the N-terminal, followed by several hydrophobic amino acid residues essential for HSF oligomerization, basic amino acid residues for nuclear localization, the nuclear export signal (NES) at the C-terminus and short peptide motifs for transcriptional activation of HSFs. Shuttling of HSFs between the cytoplasm and the nucleus is regulated by the nuclear localization signal and NES at the C-terminus of class A HSFs. Activation motifs or aromatic and hydrophobic amino acid residues (AHAs) located proximal to the NES are crucial for transcriptional activation via interaction with other transcriptional elements. Transcriptional, posttranscriptional, and protein modifications give rise to the structural and functional diversification of HSFs [8].

At normal temperatures, HSPs expressed at basal levels bind to HSFs to prevent them from activating the expression of heat-responsive genes and increasing the transcription of various HSP-encoding genes [9]. When stimulated by heat stress, denatured and misfolded proteins are sensed by HSPs. Misfolded proteins bind to HSPs, thereby releasing HSFs for activation of heat stress responses. Transcriptional profiling of HSFs revealed their critical roles in the upregulation of other stress-inducible genes under various biotic/abiotic (pathogen, wound, cold, drought, heavy metal, light, pH, oxidative, salinity) stresses and developmental/physiological (flowering, apoptosis, circadian rhythm) conditions. The overlapping of different stress response profiles suggests intricate crosstalk in multiple networks and pathways. In-depth reviews of the structure and function of HSFs have been conducted [10,11].

The class A members of HSFs mainly function to activate HSRs [12,13]. In primary HSRs, HSFA1s (including HSFA1a, HSFA1b, HSFA1d, and HSFA1e), HSFA2, and HSFA3 play pivotal roles as class A members [14]. HSFA1s serve as the most important transcriptional activators of HSR gene expression [15,16]. HSFA1s function as master regulators of HS-regulated gene expression in plants because they can further bind to other TFs [17,18]. For instance, under ambient temperature, HSP70 and HSP90 repress HSFA1 activity through direct protein interactions. The interaction negatively regulates HSFA1 activity by repressing transcription activation and nuclear localization [19]. Under HS, direct protein interactions activate a series of downstream target genes, including DEHYDRATION-RESPONSIVE ELEMENT BINDING 2A (DREB2A), HSFA2, HSFA7A, HSFA7B, and MBF1C, releasing these TFs from the cytoplasm to nuclei for transcription activation and expression of heat-induced genes [19]. HSFB1, HSFB2A, and HSFB2B can also be induced by HSFA1. Notably, HSFB1 and HSFB2B function as transcriptional repressors. Negative regulation of HSFA2 and HSFA7A expression acts as a modulator of the action of class A HSFs [20]. Interestingly, a recent study reported that by investigating genome-wide chromatin changes related to the transcriptional reprogramming response to HS in tomato, HSFA1a-mediated chromatin reorganization likely drives the expression of HS-responsive genes [21]. Heat stress induces rapid changes in chromatin architecture, leading to the transient formation of promoter enhancer contacts, thereby inducing HSRs [21]. In addition, the activity of HSFA1 is tightly controlled by protein modification. This explains why the impact of the overexpression of HSFA1 on the upregulation of HS-inducible genes is relatively limited [19,22] compared with the overexpression of other HS-inducible HSFAs, such as HSFA2 and HSFA3 [14,19,23,24]. Posttranslational regulation, such as phosphorylation and SUMOylation, plays a major role in the regulation of HSFA1 activity. For example, it has been reported that two protein kinases, CYCLIN-DEPENDENT KINASE A1 (CDKA1) and CALMODULIN-BINDING PROTEIN KINASE 3 (CBK3), can facilitate the DNA-binding ability of HSFA1 by interacting with HSFA1 in Arabidopsis [25,26]. Another phosphatase, PROTEIN PHOSPHATASE 7 (PP7), is also known to associate with HSFA1, although how it functions to mediate HSFA1 activity is unclear [27]. Other regulators have been nicely reviewed by Ohama et al. [28].

To date, the understanding of class C HSFs remains unclear. The highly conserved DNA-binding domains of class C HSFs imply strong evolutionary pressure for functional conservation. In contrast, variabilities in the OD and AHA domains encourage functional divergence of HSFs. For instance, the expanded HSFC family observed in monocots is hypothesized to carry out regulatory activities in response to stressors, developmental processes, and fine-tuning of gene expression [29].

### 2.2. Other Transcriptional Regulators

Other TF families that also play roles in the HS response include the DREB2A, DREB2C, MULTIPROTEIN BRIDGING FACTOR 1C (MBF1C), NAM/ATAF1/2/CUC2 (NAC), WRKY, basic REGION/LEUCINE ZIPPER (bZIP), and MYB families (Figure 1). Increasing amounts of data are indicating that the AP2/ERF family DREB2A is another important regulator under HS [30]. As part of HSFA1 activation, the expression of DREB2A is stimulated [14]. Then, DREB2A directly activates HSFA3 expression which functions as a heat-induced gene in acquiring thermotolerance and/or heat stress resistance. Moreover, VASCULAR PLANT ONE-ZINC-FINGER PROTEIN (VOZ1) from the NAC family inhibits the activity of DREB2C, thereby abolishing the downstream activities of HSFA3 [31]. Interestingly, DREB2A functions in the crosstalk between heat and drought stress signaling, indicating its specific roles in abiotic stress. DREB2A can be induced by JUNGBRUNNEN 1 (JUB1) and MBF1C in addition to HSFA1s under heat stress [32,33,34]. In contrast, the expression of DREB2A is repressed by GROWTH-REGULATING FACTOR 7 (GRF7) under nonstress conditions [35]. Posttranslational regulation processes, such as phosphorylation and SUMOylation, also play important roles in the regulation of DREB2A activity [28].

NUCLEAR FACTOR-Y (NF-Y) family members are central regulators of HS-responsive transcription in plant cells as well [36]. A recent study showed that the SUMO E3 ligase SAP AND MIZ1 DOMAIN-CONTAINING LIGASE1 (SIZ1) interacts with NF-YC10 and enhances its SUMOylation during HS. The SUMOylation of NF-YC10 facilitates its interaction with, and the nuclear translocation of, NF-YB3. After translocation to the nucleus, the NF-YC10/NF-YB3 dimer binds to NF-YA2 to form an NF-YC trimeric complex to promote the transcription of HS-responsive genes, thereby improving heat tolerance in plants [36]. Moreover, a previous study reported that NF-YB3 translocates to the nucleus and couples with DNA POLYMERASE II SUBUNIT B3-1 (DPB3-1) to form a trimer with NF-YA2, thereby promoting DREB2A activity under heat stress [37,38].

NAC transcription factors influence several pathways, including ROS, HSFBS, HSFA1s, and DREB2A. Mechanistically, they bind to promoters of HSFs, thereby increasing the expression of heat-induced genes and mounting the response against heat stress [39]. The influences of WRKY, MYB, and bZIP on the HS regulatory pathways are minimal relative to NAC. TFs of the WRKY family positively regulate HS regulatory pathways. MYB30 regulates the HS response through ANNEXIN (ANNs) [40]. Binding of MYB30 to ANNs represses their expression, thereby modulating cytosolic calcium ion concentrations. In times of HS, the activity of MYB30 is inhibited, enabling the downstream responses of ANNs, which lead to an influx of cytosolic calcium concentration, eliciting the HS response [40]. bZIP28 and bZIP60 positively regulate the expression of heat-induced genes. IRE1 HS-dependently splices bZIP60 RNA, leading to the production of bZIP60 protein lacking the transmembrane domain and translocates it into the nucleus for heat-induced gene expression [41]. On the other hand, binding between BINDING PROTEIN (BiP) and bZIP28 inactivates actions by bZIP28 under nonstress conditions [42].

It is interesting to note that the circadian clock proteins REVEILLE 4 (RVE4) and RVE8 have been found to positively impact thermotolerance by activating HSR gene expression [43].

### 2.3. Epigenetic Regulation in the HSRs

Multilayered regulatory systems under HS include epigenetic regulation [28]. Epigenetic regulation—involving DNA methylation, histone modification, and chromatin remodeling—is stimulated upon HS. The molecular mechanisms underlying AT have been studied and are mainly related to epigenetic regulation, especially histone modification, which modulates transcriptional activity through either “open” or “closed” chromatin configurations [44]. Furthermore, histone modifications function in AT by exploiting the involvement of TFs.

Multiple studies have indicated that HSFA2 is a master regulator of AT [45,46] (Figure 2). The maintenance of HSP gene expression by HSFA2 prolongs AT. One recent study showed that HSFA2 recruits histone methyltransferases to memory loci and leads to deposition of di- and trimethylation of histone H3 lysine 4 (H3K4me2/3). This kind of histone modification induces sustained expression of various types of heat memory genes, which are specifically required for AT in Arabidopsis [47]. After a nonlethal priming HS, H3K4me2/3 hallmarks the memory loci as recently transcribed loci, enabling these memory loci to be rapidly or more strongly re-induced (i.e., refer to a kind of transcriptional memory) by subsequent lethal HS [47]. Furthermore, it has been reported that HSFA2 can assemble into a heteromeric HSF complex alongside HSFA3/FORGETTER3 (FGT3), which subsequently triggers the expression of memory genes, thereby orchestrating AT [46]. However, it is unclear what distinguishes memory from nonmemory genes. A recent study identified the global target genes of these two key memory HSFs, HSFA2 and HSFA3, using time course Chromatin Immunoprecipitation-sequencing (ChIP-seq), and HSFA2 and HSFA3 showed nearly identical binding patterns [47]. In vitro and in vivo binding strength is highly correlated, indicating the importance of DNA sequence elements. In particular, genes with transcriptional memory are strongly enriched for a tripartite heat shock element (HSE) and are hallmarked by three features: low expression levels under normal conditions, accessible chromatin environment, and heat stress-induced enrichment of H3K4 trimethylation [48].

Furthermore, apart from FGT3, other FGTs also function in AT from various physiological aspects. FGT2 encodes a protein phosphatase of type 2C (PP2C) and interacts with phospholipase Dα2 (PLDα2) [18]. The absence of functional FGT2 or PLDα2 results in an impairment of AT, suggesting that alterations in the composition of phospholipids in the plasma membrane may play a critical role in AT [18]. The deposition of activation marks might be mediated by the COMPASS-like complex through stress-specific TFs such as HSFA2 and HSF3 to memory genes. 

Unlike H3K4me activation marks, PRC2-catalyzed H3K27me3 histone marking of target genes usually transcriptionally represses gene expression, which is mediated by conserved JUMONJI (JMJ)-domain-containing histone demethylases [49,50]. Four JMJ H3K27 demethylases have been reported to mediate the removal of H3K27me3 histone marks in the heat memory gene locus during heat memory, comprising JMJ30, JMJ32, EARLY FLOWERING 6 (ELF6), and RELATIVE OF EARLY FLOWERING 6 (REF6) in Arabidopsis [50]. These four demethylases catalyze sustained removal of H3K27me3 histone marks at two memory loci, HSP22 and HSP17.6C, and thus facilitate enhanced reinduction of these genes in response to subsequent HS [50]. In addition to H3K4 methylation and H3K27 demethylation, histone acetylation is associated with active gene expression. The evolutionarily conserved histone chaperone ANTISILENCING FUNCTION 1 (ASF1) is associated with the removal of histone H3 lysine-56 acetylation (H3K56ac) through the deposition of unmodified histone H3 [51].

AT is typically somatic and lasts for a short time (several days in *Arabidopsis* seedlings) [52,53], but long-term transgenerational AT occurs (though rarely). A regulatory loop of HSFA2-REF6 has been reported to mediate transgenerational AT at some loci in *Arabidopsis* [54]. During prolonged HS exposure, HSFA2 directly activates the expression of REF6 and the SWI/SNF family chromatin remodeler BRAHMA (BRM). REF6 in turn removes H3K27me3 at HSFA2 and recruits BRM to further upregulate HSFA2 expression, leading to the induction of a set of HSRs [55]. The positive feedback loop of HSFA2-REF6/BRM is maintained after heat exposure and further transmitted maternally to the immediate progeny, which may promote transgenerational adaptation to HS [54]. Moreover, a retrotransposon called ONSEN is also a target of HSFA2 in addition to HSFA1 during HS, indicating that it plays an important role not only in HSR but also in AT. Previous research has reported that HSFA2 stimulates ONSEN, which is associated with H3K4me methylation and H3K27me3 demethylation, thus forming a feedback loop with REF6. As a result, these histone modifications confer an HS response and transgenerational heat memory in plants [28].

Nucleosome dynamics through chromatin remodeling are also related to AT. The helicase FGT1 associates with the ATP-dependent chromatin remodelers SWI/SNF (BRM) and Imitation Switch (ISWI) (CHROMATIN-REMODELING PROTEIN 11 (CHR11) and CHR17) to reduce nucleosome abundance at memory loci in the phase of memory after priming HS [55]. The removal of nucleosomes facilitates the sustained activation of memory genes and thus confers AT in *Arabidopsis* [55].

In summary, the chromatin-based histone modification for AT regulation primarily encompasses HSFA2-mediated H3K4 hyper-trimethylation, JMJ demethylase-dependent H3K27 demethylation, ASF1-mediated H3K56ac, and FGT1-BRM/CHR11/CHR17-dependent nucleosome positioning, and these mechanisms collectively contribute to the modulation of AT in land plants [49]. The effect of AT is usually erased in several days. The precise mechanism underlying the deposition and removal of these histone-modifying proteins in the process of AT remains to be understood.

### 2.4. Regulatory RNA (microRNA, siRNA, lncRNA, and circRNA) 

Much involvement of regulatory RNA has been documented when plants have been subjected to heat stimulus (Figure 1). The regulatory activities of TFs or genes are attributed to noncoding RNAs, including microRNAs, small interfering RNAs (siRNAs), long noncoding RNAs (lncRNAs), and circular RNAs (circRNAs). MicroRNAs are small noncoding RNAs that regulate target messenger RNA (mRNA) through degradation or translational repression of mRNA. Gene expression is negatively regulated, thereby inhibiting translation. miR398 acts downstream of HSFA7b and HSFA1s during HS [56]. When faced with oxidative stress, the activity of miR398 is repressed, leading to the accumulation of the COPPER/ZINC SUPEROXIDE DISMUTASE 1 (CSD1) and CSD2 genes. Encoded closely related Cu/Zn superoxide dismutases detoxify superoxide radicals, contributing to oxidative stress tolerance [56]. miR156 is one of the targets of noncoding RNAs when induced by heat stress. Its regulatory activity in the heat stress response is exerted via the repression of SQUAMOSA-PROMOTER BINDING-LIKE (SPL) TFs, including SPL2, SPL9, SPL11, and SPLs, which act as upstream repressors of HSFA2 [57]. When faced with heat stimulus, actions by miR156 leave sustained expression of HSFA2 even after recovery from HS, adding AT to plants’ defenses for combating future encounters with HS. Additionally, miR159 and miR396 were reported to target MYB and WRKY, respectively, in conferring plant thermotolerance [58].

A recent study revealed that miR165/166 and its target transcript, PHABULOSA (PHB), can form a module to regulate HSFA1 at the transcriptional and translational levels in response to HS [59]. On the one hand, under normal conditions, PHB directly represses HSFA1 transcription and globally regulates the expression of HS-responsive genes. During HS, the accumulation of miR165/166 is triggered, leading to the downregulation of PHB, thereby releasing HSFA1 from the HSFA1/PHB complex to induce the expression of HS-responsive genes. On the other hand, the lack of PHB induces the transcription of HSFA1s and HSFA2 in response to HS [59].

As mentioned above, ONSEN responds to HS as a target of HSFA1. Moreover, a siRNA-mediated pathway regulates ONSEN activity. Inhibitory action by siRNA on ONSEN yields upregulation of heat-induced gene expression. Apart from interaction with ONSEN, HSFA1s induce thermotolerance by binding at promoters, triggering transcriptional activation of the HEAT-INDUCED TAS1 TARGET 1 (HTT1) and HTT2 genes. In addition to the influence of HSFA1s, the expression of HTT1 and HTT2 genes is also garnered by trans-acting siRNA (TAS1). The natural antisense transcript siRNA (nat-siRNA) contributes to heat resistance by negatively regulating HTT1 and HTT2. CircRNAs are single-stranded RNAs that join head-to-tail in a circular form. They are reported to play a role in the regulation of the plant HS response, synergistic with plant hormone signal transduction [60].

## 3. Other Cell Activities for Adaptation to HS

Exposure of plant cells to high temperatures results in cellular damage and can even lead to cell death. The damage can be ascribed to the action of misfolded proteins and reactive oxygen species (ROS), which accumulate during abiotic stresses such as heat stress. To protect cells, protein homeostasis and ROS homeostasis are essential. Thus, we briefly review the recent study below about how plants maintain protein homeostasis and ROS homeostasis to cope with HS downstream of transcriptional regulation (Figure 3).

### 3.1. Protein Homeostasis

HS can induce the production of unfolded proteins in plant cells, which can be cytotoxic. To survive HS, plants must undertake mechanisms to either renature or degrade these unfolded proteins. HSPs (or chaperones) assume vital functions in preserving protein quality by facilitating the renaturation of denatured proteins during HS [61]. These chaperones actively participate in diverse cellular processes, encompassing the folding of nascent proteins on ribosomes and facilitating transport across membranes, besides preventing cellular damage from encountering various stressful conditions [62]. The coordinated interaction between chaperones and proteases is commonly referred to as ‘protein quality control’ [62]. In Arabidopsis, the main families of plant chaperones are HSP60, HSP70, HSP90, HSP100, and small HSPs (sHSPs), all of which have been reported to confer thermotolerance [61]. For example, HSP100 enhances heat tolerance by resolubilizing protein aggregates in Arabidopsis [63]. Moreover, OsHSP101 is a positive regulator of thermotolerance and heat memory in rice [64]. In addition to HSPs, the 26S proteasome α2 subunit protein THERMOTOLERANCE (OsTT1) removes heat-induced cytotoxic denatured proteins, thus enhancing thermotolerance in rice [65].

Autophagy is another homeostasis pathway that instrumentally regulates plant adaptations to HS by removing nonfunctional proteins and damaged cellular components [66]. Neighbor of BRCA1 (NBR1)-mediated selective autophagy can either target ubiquitinated protein aggregates to enhance basal heat tolerance or contribute to the selective degradation of HSPs after HS to reduce AT [67,68,69]. In canonical autophagy, several core autophagy-related (ATG) proteins are recruited hierarchically to form double-membrane organelles termed autophagosomes [70]. When autophagy is impaired in Arabidopsis and tomato plants, aggregated proteins tend to accumulate, resulting in reduced heat stress tolerance [68,71]. ATG8 homologs in mammals have also been shown to target single-membrane compartments such as phagosomes and endosomes. A recent study in plants reported a noncanonical function of ATG8 in regulating the restoration of the Golgi apparatus damaged by HS. Short-term acute HS causes vacuolation of the Golgi apparatus and translocation of ATG8 to the dilated Golgi membrane. The inactivation of the ATG conjugation system, but not of the upstream autophagic initiators, abolishes the targeting of ATG8 to the swollen Golgi, causing a delay in Golgi recovery after HS. Using TurboID-based proximity labeling, CLATHRIN LIGHT CHAIN 2 (CLC2) was identified as an interacting partner of ATG8 via the ATG8-interacting motif (AIM) – LIR/AIM docking site (LDS) interface. CLC2 is recruited to the cisternal membrane by ATG8 to facilitate Golgi reassembly [66].

### 3.2. ROS Homeostasis

Exposure to high temperatures also quickly enhances the production of ROS molecules such as hydrogen peroxide (H_2_O_2_), superoxide (O_2_^•−^), singlet oxygen (^1^O_2_), and hydroxyl radical (^•^OH), which act as signaling molecules for plants to adapt to stress conditions [72]. The function of ROS in peroxisomes is vital in promoting plant survival through the initiation of plant reproduction. However, the accumulation of ROS causes oxidative damage to lipids and DNA, which triggers calcium influx leading to apoptosis. Under heat stress, PM-localized NADPH oxidases (known as RESPIRATORY BURST OXIDASE HOMOLOG [RBOH] in plants and NADPH oxidase [NOX] in animals) appear to be the primary source of ROS production, as ROS accumulation has been successfully blocked by an NADPH oxidase inhibitor [73]. Arabidopsis atrbohB and atrbohD mutants exhibit reduced thermotolerance [74], indicating that plants need to maintain ROS homeostasis to attain thermotolerance. Moreover, H_2_O_2_ also functions partly as a signaling molecule as a consequence of the secondary stress response, where the accumulation of H_2_O_2_ triggers upregulation of HSFs.

To reduce the effects of oxidative damage, plants produce antioxidants to combat the accumulation of ROS to prevent oxidative damage [75]. Ascorbate peroxidases (APXs) and catalases (CATs) are two types of ROS-scavenging enzymes that have been reported to detoxify ROS [76]. In wheat, it was found that melatonin improves the activity of the antioxidant enzymes superoxide dismutase (SOD), catalase (CAT), and peroxidase (POD), which enhances the heat tolerance of the plant.

The activation of ROS signaling through HSFs has been supported by evidence, which includes findings on the interaction between HSFs and genes involved in ROS scavenging. Notably, the expression of APX1 has been shown to be regulated by HSFA2; overexpression of HSFA2 led to elevated APX1 expression, whereas AtHSFA2 knock-out mutants exhibited reduced APX1 expression [77]. Accordingly, AtHSFA2 overexpression lines showed increased heat and oxidative stress tolerance [77]. Expressing a dominant-negative construct for AtHSFA4a in Arabidopsis has been shown to impede the accumulation of APX1 transcripts [78,79]. Notably, the AtHSFA4a dominant-negative construct not only hampers the accumulation of APX1 transcripts but also inhibits the accumulation of the H_2_O_2_-responsive zinc-finger protein ZINC FINGER OF ARABIDOPSIS THALIANA12 (ZAT12), which is essential for APX1 expression under oxidative stress conditions. The ZAT12 promoter contains HSE binding sites [80]. Therefore, HSFA4a might directly associate with the ZAT12 promoter [79]. However, the presence of HSEs in the promoter region of the APX1 gene itself suggests the possibility of direct activation via heat shock factors (HSFs) [81,82]. In addition, previous studies have shown that expression of heat-responsive genes is increased upon application of the ROS H_2_O_2_. For example, AtHSP17.6 and AtHSP18.6 exhibit similar expression levels following H_2_O_2_ application at normal conditions as they do following HS [83].

In conclusion, heat directly induces HSF activity but also indirectly induces HSF activity via ROS signaling. HSFs, in turn, affect the expression of HSP/ROS scavenger genes. According to multiple studies, these contrasting effects allow for boosting HSRs at the very onset of stress while preventing subsequent oxidative damage [84].

## 4. Application of Heat Stress-Related Transcription Factors

### 4.1. TF Modulation to Improve Plant Resilience

To date, although most research on elucidating the mechanisms of HSFs has been performed in the eudicot Arabidopsis, it is reported that there are 25 HSFs in maize, rice, and sorghum, and that they are evolutionarily conserved in seed plants [85]. Translational research on crops is needed to demonstrate the role of TF in modulating crop performance in times of HS. This is particularly critical as the world faces the harsh reality of climate change, and food demand rises to feed an estimated 9 billion population in 2050 [3]. 

With increasing developments in gene discovery, genetic modification has been employed for crop improvement. Previous studies have reported that introducing HSF and HSP enhances thermotolerance in crops; for example, overexpression of tomato SlHsp21 or soybean GmHsfA1d promotes tolerance to HS in tomatoes and soybeans, respectively [86,87]. However, the overproduction of active proteins often gives rise to growth retardation under nonstress conditions [88]. A new strategy for crop improvement utilized an inducible promoter to prevent the negative effects on growth under nonstress conditions [88]. Optimized and more sensitive promoters from HS-inducible genes can regulate the expression levels of such active factors. In addition, the discovery that plants have ‘cognitive abilities’ to acquire memories based on epigenetic regulation provides another strategy to enable an intelligent design for crop improvement. Recently, Oberkofler and Baürle reported an inducible system for epigenome editing in Arabidopsis thaliana using a heat-inducible dCas9 to target a JUMONJI (JMJ) histone H3 lysine 4 (H3K4) demethylase domain to a locus of interest [89]. This system is widely applicable to the modification of histone modification levels in basic research questions and applied settings for scientists and plant breeders [89].

Since there is currently low acceptance of genetically modified (GM) crops by consumers, future research may direct the engineering of crop resilience via non-GM approaches. Given the crosstalk of TFs involved in multiple pathways, genome-wide association studies (GWAS) and quantitative trait loci may provide hints to candidate TFs for modulatory work. Some TFs have broader roles, for instance, HSFs. Genetic engineering via modulation of TFs must consider intricate networks and potential implications for other plant functions. Chemical screening using a key HSF reporter line would be another alternative strategy to identify compounds for the enhancement of crop resilience against prolonged HS, and thus crop productivity is one of the vital characteristics of food for the future [28,90].

### 4.2. Tell-Tale Signs of Abiotic Stress Using Plant Sensors

Having identified and elucidated the mechanisms of some TFs in response to HS, one potential area of translational research could be the deployment of wearable plant sensors for the detection and/or quantification of TFs/related molecules. Sensors such as these are designed to allow for early detection of any signs of abiotic stresses, including HS, facilitating necessary actions within the plant stress threshold. Miller and Mittler evaluated the feasibility of HSFs as sensors in plants based on past research works [91].

Understanding plant heat stress responses may be applied to crop resilience strategies. Hence, unraveling the crosstalk among stress sensing and early signaling pathways is necessary for the development of stress-resilient crops in the face of climate change [92].

## 5. Conclusions and Perspectives

There is an escalating threat of yield reductions in agricultural crops due to HS caused by global warming, and it is crucial to demonstrate how plants respond to HS [28]. The molecular mechanisms underlying plant responses and adaptations to HS (referred to as basal thermotolerance) and recurring HS (referred to as acquired thermotolerance (AT)) have been elucidated (Figure 4). Nonetheless, many gaps remain in our understanding of plant responses to temperature stress, especially pertaining to early signaling events and fluctuating temperatures at multiple levels [93,94]. For example, how do HS sensors transduce signals into Ca^+^, ROS, or other signaling pathways, and what exact factors are involved in these signaling cascades? Another remaining issue is how HSFA1 and DREB2A are posttranslationally activated under HS conditions or repressed under nonstress conditions, including the timing and related sites involved [28]. Although there have been a few studies performed under natural environments in the temperature stress research field [50], the majority of the studies, conducted thus far, have been performed under controlled laboratory conditions, which may not completely reflect the complexities and dynamics of the natural environment found in the field. Addressing these questions is important to successfully adapt to environmental temperature fluctuations caused by global climate change. Additionally, many studies currently focus on the mechanisms of stress tolerance at the vegetative stage of plant development. However, reproductive development and fertilization are also sensitive to stress conditions, often leading to more serious losses of crop yield [95,96,97,98]. Therefore, we need to expand our knowledge on plant stress responses to the reproductive stages of development as the related mechanisms have been recently reported [99].

## Figures and Tables

**Figure 1 ijms-24-13297-f001:**
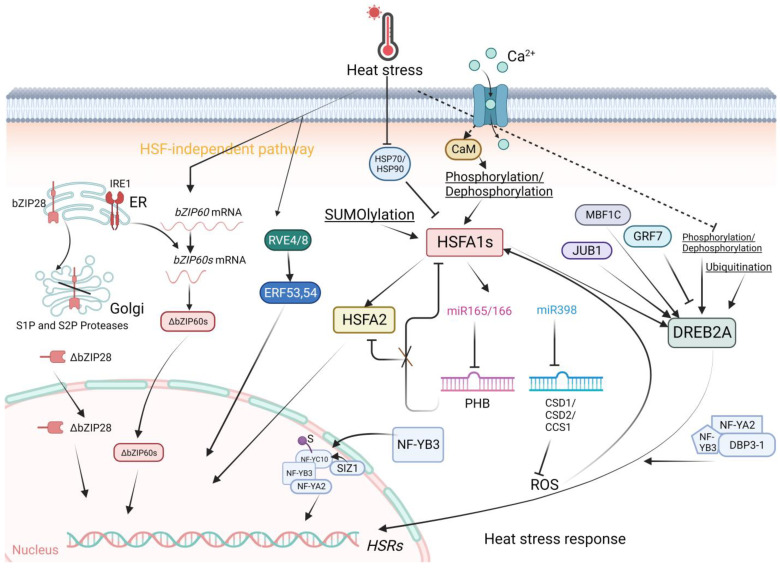
The Heat Stress Response in Arabidopsis. HSFA1s function as central regulators orchestrating plant responses to heat stress (HS). HSFA1 expression is induced by heat, and HSFA1 activity is precisely modulated by various factors. Upon heat stress, HSFA1 is released from HSP70 and HSP90, leading to HSFA1 activation. In addition, the increases of cytoplasmic Ca^2+^ levels mediated by the Ca^2+^ channels, CNGCs, triggered by HS may be important for HSFA1 activation. Post-translational modifications, such as phosphorylation/SUMOylation/Ubiquitination regulate the activity of HSFA1 and DREB2A. HSFA2 as a target of HSFA1s is an important regulator of the expression of HSR genes through sustained H3K4 methylation. The miR398 inhibits the expression of ROS scavenger genes CSD1, CSD2, and CCS1, thereby promoting ROS accumulation, which subsequently activates HSFA1s. The miR165/166–PHB module regulates thermotolerance through at least two pathways: one involving direct transcriptional regulation of HSFs, with PHB modulating HSR transcription in an HSFA1-dependent manner; and another HSFA1-independent pathway, where PHB directly regulates the transcription of heat-inducible HSFA2. Additionally, PHB physically interacts with HSFA1s, influencing their transcriptional function. DREB2A is positively regulated by MBF1C and JUB1 in response to HS, whereas it is negatively regulated by GRF7 under normal conditions. In addition, DREB2A activity is enhanced by the NF-YA2/NFYB3/DBP3-1 complex. During HS, SIZ1 facilitates the SUMOylation of NF-YC10. The SUMO conjugation on NF-YC10 enhances its association with NF-YB3 via a SUMO-SIM interaction and improves the nuclear translocation of NFYB3. In the nucleus, the NF-YC10–NF-YB3 dimer binds to NF-YA2 to form an NF-YC trimeric complex to promote the transcription of HS-responsive genes. HS triggers the ER-localized transcription factors bZIP60 and bZIP28 to translocate into the nucleus and activate HSR expression. The circadian clock proteins RVE4 and RVE8 also can induce HSR gene expression in an HSFA1-independent way. Created with BioRender.com; accessed on 31 July 2023.

**Figure 2 ijms-24-13297-f002:**
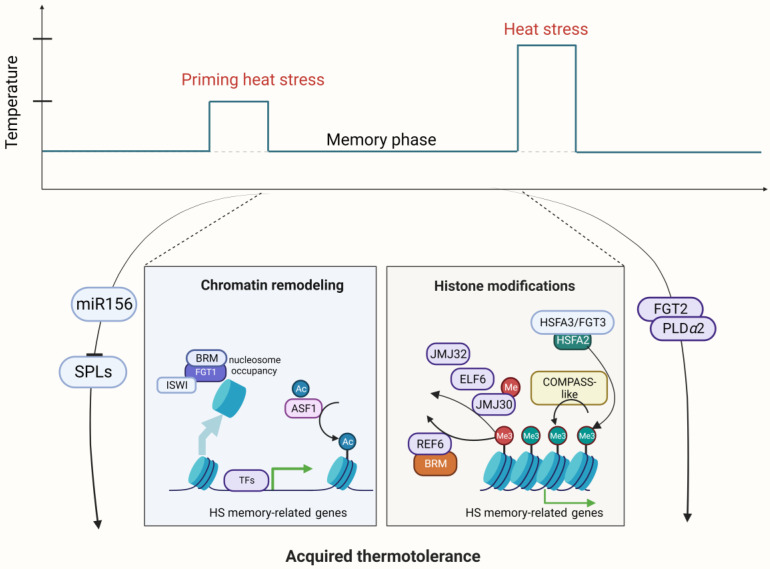
Schematic representation of regulators involved in Acquired thermotolerance (AT) in *Arabidopsis*. A mild heat stress (HS) can act as a priming cue and trigger enhanced tolerance to HS in the primed state. This primed state is maintained over time in a memory phase so that primed plants that encounter a second severe stress event are able to survive in contrast to nonprimed plants. Priming HS activates the expression of HSFA2 through HSFA1s, which are released from the HSP chaperone under priming HS and enter the nucleus. HSFA2 forms a heteromeric transcription factor complex with HSFA3, which activates the expression of HS memory genes or heat stress response (HSR) genes. Histone modifications play a critical role to regulate HS memory genes, including H3K27 demethylation by REF6, ELF6, JMJ30, and JMJ32; H3K4 trimethylation by COMPASS-like; and nucleosome positioning by the ATP-dependent chromatin remodeler complex consisting of FGT1, BRM, and ISWI. In addition, the FGT2/phosphatase-PLDa2/phospholipase complex and the miR156-SPL module are involved in the regulation of thermomemory genes in *Arabidopsis*. Created with BioRender.com; accessed on 31 July 2023.

**Figure 3 ijms-24-13297-f003:**
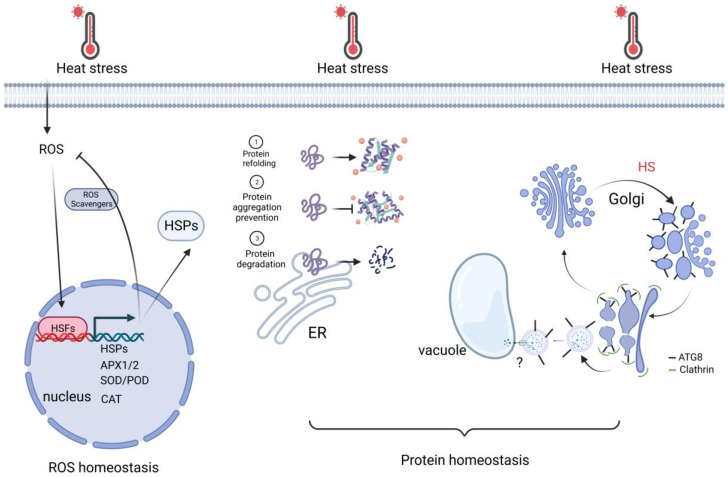
Cell activities for keeping ROS and protein homeostasis. Upon heat stress, nuclear redox oxidation is sensed by HSFs. HSFs activate HSR genes, which are important for ROS scavenging and protein homeostasis. In addition, autophagy-related (ATG) protein ATG8 is rapidly translocated to the sites of swelling Golgi bodies. It recruits the clathrin component CLC2 to mediate the vesicle budding, which fuses with the vacuole. It facilitates the reassembly of the Golgi apparatus, thereby increasing thermotolerance. Created with BioRender.com; accessed on 31 July 2023.

**Figure 4 ijms-24-13297-f004:**
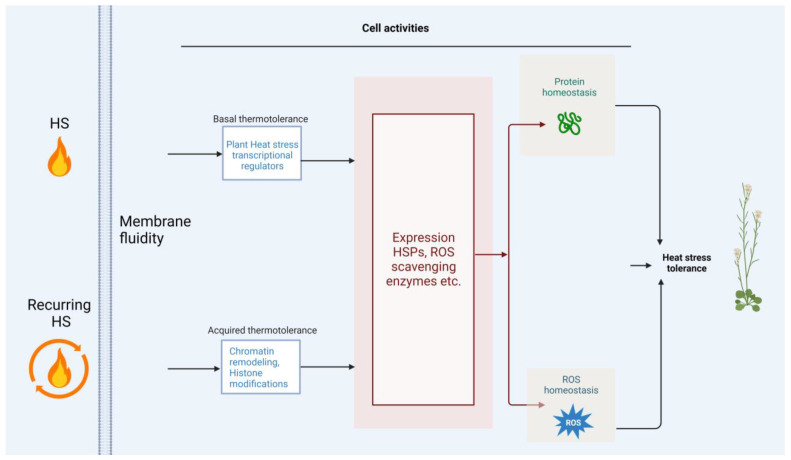
Schematic representation of how plants respond to various heat stress (HS). HS changes membrane fluidity, which may be sensed by proteins, such as Ca^2+^ channels and receptor-like kinases, localized at the plasma membrane. After plants perceive temperature signals, a set of cell activities are activated, thereby conferring plant HS tolerance. Created with BioRender.com; accessed on 31 July 2023.

## Data Availability

No new data were created or analyzed in this study.

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
