# Peer review of "Transcriptional Regulators of Plant Adaptation to Heat Stress"

_ijms, 2023, doi:10.3390/ijms241713297_

Round 1
Reviewer 1 Report
Dear authors,
I congratulate you on your work!
However, I have some small suggestions/recommendations that you can find outlined in the manuscript.
The manuscript entitled "Transcriptional Regulators of Plant Adaptation to Heat Stress" is a review of specialized literature that brings together the knowledge related to the mechanisms and physiological processes of plant adaptation/response to heat stress. It is a study addressed to plant physiology and plant breeding specialists and is of great relevance considering current and future climate changes. The originality of the manuscript is given by the grouping and synthetic presentation of the mechanisms made by the authors. From my knowledge, it is among the few articles that summarize the current information related to the subject addressed. As I specified in the suggestions made, apart from some minor recommendations/suggestions, it is a proper article.
Good thoughts!

-
Author Response
We thank you for the detailed comments. We have thoroughly revised the manuscript based on the suggestions in peer-review-31292286.v1.pdf. Primarily, we have added new references and rearranged the text.
Reviewer 2 Report
The review paper by Wang et al. summarizes recent progress in understanding the molecular mechanisms, especially transcriptional regulators, in response to heat stress in plants. The topic is timely, and the manuscript was well organized. However, I have a suggestion. I think that the manuscript could be strengthened by providing a more integrated discussion or synthesis of the cited literature. In addition, you may want to consider including in more detail one or two examples to illustrate the application of abiotic stress-related transcription factors.
See above.
Author Response
Thank you very much for your positive comments. We have strengthened the discussion and added a few sentences on the translational aspect of heat-responsive transcription factors in the revised manuscript (e.g., usage of an inducible promoter, epigenome editing, chemical library screening, etc.). We also thoroughly checked the text, removed the redundant statement, and improved the readability.
Reviewer 3 Report
Manuscript number: ijms-2560663
Manuscript title: Transcriptional Regulators of Plant Adaptation to Heat Stress
Authors: Xuejing Wang, Nicholas Tan Wui-Kiat, Fong-Yi Chung, Nobutoshi Yamaguchi, Eng-Seng Gan, and Toshiro Ito.
In the present review manuscript, an appreciable attempt is made to condense in a structured manner the most recent evidence concerning the regulation of heat stress response and adaptation in plants at the transcriptional level. The treated topic is of interest for the Journal readership, the literature coverage appears to be appropriate and an effort is made here to go beyond a mere compilation, by trying to develop conceptual frameworks which help the reader to find his/her way in a very large research field of plant science. The present manuscript is also rather well written, linguistically speaking, even if a further thorough inspection concerning clarity, editing accuracy, as well as the use of English, would be highly beneficial (see below):
- Line 33: “..other environmental issues”. Which ones? Please do not use generic sentences
- Lines 72-73: “ rendering protein‒protein interactions and trimerization during transcriptional activation”. The sentence is not clear
- Lines 84-87: “Transcriptional profiling of HSFs revealed 84 their critical roles in the upregulation of other stress-inducible genes under biotic (pathogen, wound, apoptosis, flowering, circadian clock) and abiotic (cold, drought, heavy 86 metal, light, pH, oxidative, salinity) stress conditions.” I was unaware that flowering and the circadian clock can be regarded as biotic stresses!
- Line 102: “Other HSF genes stimulated include HSFB1…”. Please correct English
- Lines 125-126: “For instance, the expanded HSFC family observed in monocots is hy-125 pothesized to carry out specialized functions [29]”. Again, the sentence is too generic…Which sspecialized functions?
- Line 132: “…also activates HSFA3 expression as a direct target gene as a heat-induced gene”. The sentence is not clear
- Lines 173-174: “facilitates its interaction with and the nuclear translocation of NF-YB3, improving the nuclear translocation of NFYB3”. This sentence appears to be tautological, in that it repeats twice just one concept, and the same.
- Line 188: check the use of the verb “to render”, in the present context. The sentence is unclear (see also lines 72-73, above, for the misuse of the verb “to render”)
- Line 191: “IRE1 splices bZIP60 into spliced bZIP60”…Another tautology here…
- Lines 195-196: add literature citation
- Line 217: spell out the acronym ChIP-seq
- Lines 242-243: that phospholipases hydrolyse phospholipids, is again a bit tautologic, in my opinion
- Lines 272-273: the sentence is not clear, which again depends on the misuse of the verb “to render” (see also lines 72-73, and line 188, above)
- Lines 324-325: I do not understand neither the meaning of this sentence, nor the way in which it is connected to the preceding period.
- Line 360: spell out the acronym NBR-1
- Line 384: spell out the acronym NOX
- Lines 390-393: again, two model samples of tautology!
- Line 417: “limited crop types”…?????...What are these?
- Line 440: never begin a sentence with a number, please. Rather, spell out the name of the cited authors, followed by the number of the citation, in square brackets
- Line 449: “The molecular and cellular mechanisms of how plants respond…”. Please correct the English style
All the above considering, I recommend major revision of the present manuscript, carefully addressing all the points raised above.
Moderate editing of English language required
Author Response
We really thank you for the detailed and important suggestions to improve the manuscript. We have incorporated all the comments. We went through the whole text and revised it for clarity and readability. We have also spelled out acronyms and double-checked references. Corrections in the revised manuscript are highlighted in yellow and the one-to-one responses to comments are as follows:
- Line 33: “..other environmental issues”. Which ones? Please do not use generic sentences
Response: Thanks for your nice suggestion. We have removed the confusing statement in the revised manuscript.
- Lines 72-73: “rendering protein‒protein interactions and trimerization during transcriptional
activation”. The sentence is not clear
Response: Sorry for the ambiguous description. We have elaborated the functional domains of HSFs in detail.
- Lines 84-87: “Transcriptional profiling of HSFs revealed 84 their critical roles in the upregulation
of other stress-inducible genes under biotic (pathogen, wound, apoptosis, flowering, circadian
clock) and abiotic (cold, drought, heavy metal, light, pH, oxidative, salinity) stress conditions.” I
was unaware that flowering and the circadian clock can be regarded as biotic stresses!
Response: Thank you for pointing these errors. We have re-listed various biotic/abiotic stress and developmental/physiological conditions.
- Line 102: “Other HSF genes stimulated include HSFB1…”. Please correct English
Response: We have corrected it in the revised manuscript.
- Lines 125-126: “For instance, the expanded HSFC family observed in monocots is hypothesized to carry out specialized functions [29]”. Again, the sentence is too generic…Which is specialized functions?
Response: We have described in more detail in the revised manuscript.
- Line 132: “…also activates HSFA3 expression as a direct target gene as a heat-induced gene”.
The sentence is not clear
Response: We have corrected it in the revised manuscript, by splitting them into two sentences.
- Lines 173-174: “facilitates its interaction with and the nuclear translocation of NF-YB3, improving the nuclear translocation of NFYB3”. This sentence appears to be tautological, in that it repeats twice just one concept, and the same.
Response: We removed the redundant statement.
- Line 188: check the use of the verb “to render”, in the present context. The sentence is unclear
(see also lines 72-73, above, for the misuse of the verb “to render”)
Response: We have corrected them in the revised manuscript, changing from “to render” to ”enabling”
- Line 191: “IRE1 splices bZIP60 into spliced bZIP60”…Another tautology here…
Response: We have corrected it in the revised manuscript.
- Lines 195-196: add literature citation
Response: We have added the reference.
- Line 217: spell out the acronym ChIP-seq
Response: We have spelled out “Chromatin Immunoprecipitation-sequencing (ChIP-seq)”.
- Lines 242-243: that phospholipases hydrolyse phospholipids, is again a bit tautologic, in my
opinion
Response: We removed “phospholipases hydrolyse phospholipids”
- Lines 272-273: the sentence is not clear, which again depends on the misuse of the verb “to
render” (see also lines 72-73, and line 188, above)
Response: We have corrected them in the revised manuscript.
- Lines 324-325: I do not understand neither the meaning of this sentence, nor the way in which it
is connected to the preceding period.
Response: We are sorry for the confusing description. We removed the sentence since it does not make sense.
- Line 360: spell out the acronym NBR-1
Response: We have corrected it in the revised manuscript. “Neighbor of BRCA1 (NBR1)-mediated”
- Line 384: spell out the acronym NOX
Response: We have spelled out “NADPH oxidase [NOX]”
- Lines 390-393: again, two model samples of tautology!
Response: We removed the sentence.
- Line 417: “limited crop types”…?????...What are these?
Response: We have specified crop species used in the previous studies.
- Line 440: never begin a sentence with a number, please. Rather, spell out the name of the cited
authors, followed by the number of the citation, in square brackets
Response: We have corrected it in the revised manuscript.
- Line 449: “The molecular and cellular mechanisms of how plants respond…”. Please correct the English style
Response: We have corrected it in the revised manuscript.
Round 2
Reviewer 3 Report
I found satisfactory the replies provided by the Authors to the comments I raised on their original submission.